# The Role of PI3K/AKT and MAPK Signaling Pathways in Erythropoietin Signalization

**DOI:** 10.3390/ijms22147682

**Published:** 2021-07-19

**Authors:** Zuzana Tóthová, Martina Šemeláková, Zuzana Solárová, Jana Tomc, Nataša Debeljak, Peter Solár

**Affiliations:** 1Institute of Medical Biology, Faculty of Medicine, P.J. Šafárik University in Košice, 04001 Košice, Slovakia; zuzana.tothova1@student.upjs.sk (Z.T.); martina.semelakova@upjs.sk (M.Š.); 2Institute of Pharmacology, Faculty of Medicine, P.J. Šafárik University in Košice, 04001 Košice, Slovakia; zuzana.solarova@upjs.sk; 3Medical Centre for Molecular Biology, Institute of Biochemistry and Molecular Genetics, Faculty of Medicine, University of Ljubljana, SI-1000 Ljubljana, Slovenia; jana.tomc@mf.uni-lj.si (J.T.); natasa.debeljak@mf.uni-lj.si (N.D.)

**Keywords:** erythropoietin, PI3K, MAPK

## Abstract

Erythropoietin (EPO) is a glycoprotein cytokine known for its pleiotropic effects on various types of cells and tissues. EPO and its receptor EPOR trigger signaling cascades JAK2/STAT5, MAPK, and PI3K/AKT that are interconnected and irreplaceable for cell survival. In this article, we describe the role of the MAPK and PI3K/AKT signaling pathways during red blood cell formation as well as in non-hematopoietic tissues and tumor cells. Although the central framework of these pathways is similar for most of cell types, there are some stage-specific, tissue, and cell-lineage differences. We summarize the current state of research in this field, highlight the novel members of EPO-induced PI3K and MAPK signaling, and in this respect also the differences between erythroid and non-erythroid cells.

## 1. EPO and Erythropoiesis

Erythropoiesis is the process of creating mature red blood cells from pluripotent hematopoetic progenitor cells, resulting in the production of approximately two million erythrocytes per second [1]. Erythropoietin (EPO) leads through the erythropoiesis to an increased number of red blood cells to provide the oxygen supply for organs and tissues. Low-level secretion of EPO is sufficient for classical steady-state erythropoiesis but its production is increased during hypoxia and anemia. The first site of EPO production is considered to be neural crest cells and neuroepithelial cells that transiently secrete EPO during the early stages of embryogenesis [2]. In the late embryogenesis, EPO distribution is shifted to the fetal liver [3], whereas adult EPO production resides in the specialized kidney interstitial fibroblasts called pericytes [4]. Erythrocytes in mammals are derived from megakaryocyte-erythroid progenitors differentiating into burst-forming unit-erythroid (BFU-Es) and then to colony-forming unit-erythroid (CFU-Es) [5]. Subsequently, arising erythroid lineage differentiates from the proerythroblast stage, followed by basophilic, polychromatophilic, and orthochromatic erythroblast with a gradually decreased number of EPOR. Beginning from BFU-E stage and ending in the orthochromatic phase, EPO accompanies differentiating cells throughout their maturation and induces transcriptional reprogramming [6]. The final product of erythropoiesis is red blood cells with its characteristic biconcave shape ideally adapted for transport of respiration gases. Furthermore, in mature red cells most of organelles are lacking to ensure the biggest volume for hemoglobin accumulation.

Production of mature red blood cells is a sophisticated multistep process located in erythroblastic islands, a specialized niche within the bone marrow with a population of erythroid cells in different stages of differentiation and the central macrophage [7] that is a source of regulatory cytokines for red blood cells’ maturation such as insulin-like growth factor-1 (IGF-1), bone morphogenetic protein-4 (BMP-4), and ferritin [8,9,10]. However essential, the main regulatory molecules for erythroid cell production are EPO and its receptor EPOR. Numerous hormones, cytokines, and members of extracellular matrix participate in the regulation of erythropoiesis, for example, in stem cell factor SCF, IGF-1, corticosteroids, interleukines IL-3 and IL-6, fibronectin, hepcidin, or erythroferrone. The effect of these molecules orchestrate intracellular transcriptome, epigenetic, and proteome changes in differentiating erythroid cells. The major EPO activated signaling pathways differentially influence transcriptome reprogramming of the hematopoietic stem cells and their cell lineage commitment. In addition, regulation of erythroid proliferation strongly depends on the protein abundance of AKT and MAPK pathway mediators [11].

As we have described the role of the STAT5 protein as a key molecule of EPO signaling in our recent paper [12], we will now focus on the role of the PI3K and MAPK pathways in both erythroid and non-erythroid cells in more detail.

## 2. EPOR/PI3K/AKT and Erythropoiesis

The EPOR/PI3K signaling cascade is crucial in mediating signals for survival, proliferation, and terminal differentiation of erythroid progenitors [5,13]. PI3K is composed of one regulatory subunit (P85) and one catalytic subunit (P110). P85 subunit is associated with the tyrosine residues (Y607 and Y508 sites) of EPOR and activates the PI3K pathway [14]. After binding of EPO to its receptor, the p85-α subunit has an essential role for EPOR endocytosis in addition to the ubiquitination of Epsin1 and Cbl proteins [15]. Activated PI3K phosphorylates secondary messenger molecules are known as phosphoinositide 3,4 bisphosphate PI(3,4)/P2 and phosphoinositide 3,4,5-trisphosphate PI(3,4,5)/P3 [16]. Activation of phosphoinositide-dependent kinase 1 (PDK1) leads to the phosphorylation of a serine-threonine kinase AKT (known also as protein kinase B), considered as a central mediator of PI3K/AKT signaling. Among successful signal transduction, the nuclear translocation of AKT is required for EPO-induced erythroid differentiation [17]. Subsequently, downstream targets of PI3K/AKT include transcription factors FOXO3 and GATA-1 that are essential for normal erythroid cell development [18,19], hypoxia-induced factor-1α (HIF1α) [20,21], and the mammalian target of rapamycin (mTOR) [11]. Furthermore, negative regulators such as phosphatase, tensin homologue (PTEN), and carboxyl-terminal modulator protein C can block the PI3K/AKT signaling pathway [22] (Figure 1). EPO-mediated PI3K/AKT signaling was mapped to identify EPO responsive genes in the CD34+ cells, the earliest detectable erythroid progenitors of human bone marrow. Gene expression profiling identified a plethora of target genes from which the most significant were: GNG2 and RDS20 involved in G-protein signaling, and PABPN1and CPSF5 required for progressive and efficient polymerisation of poly(A) tails on the 3′ ends of a eukaryotic gene. Moreover, mitotic regulators ANAPC4 and SEMA3-F involved in Semaphorin signaling and KIAA0746 in NOTCH signaling were detected in addition to the negative regulator of JAK kinases PTPRC. Furthermore, cell cycle progression and differentiation were controlled via EPO/PI3K hyperphosphorylation of the RB protein and upregulation of cyclin D_3_, E, and A in addition to the upregulation of specific markers of immature erythroid progenitors c-KIT and E-cadherin [23]. Interestingly, several genes associated with transcriptional repressions such as THG-1, KLF8, and CNOT3 were represented among the downregulated genes. Indeed, Sivertsen et al. [23] proved that most of the EPO responsive genes are regulated by PI3K-dependent fashion apart from CISH and PIM1 that are known to be inducible by STAT5.

## 3. EPOR/MAPK and Erythropoiesis

During erythroid maturation, MAPK as activated by EPO plays an important role in myeloid gene suppression, while the PI3K pathway is critical for erythroid gene induction [24]. Recently, a wide proteome study exhibited changes across differentiation from early erythroid progenitors into nearly mature erythrocytes [25]. Although the MAPK pathway is predominantly c-Kit driven and downregulated before the loss of EPOR/JAK2 activity, it is important for the early stages of erythropoiesis and for terminal maturation delay. As the kinetic of MAPK protein expression varies across the stages of erythroid maturation, it is assumed that its activity may be suppressed post-translationally. These findings are supported by other earlier studies in which downregulation of MAPK signaling promotes terminal maturation [26].

MAPKs are a family of evolutionarily conserved Ser/Thr kinases that play crucial roles in proliferation, migration, differentiation, senescence, and apoptosis. They are also frequently connected and/or involved in the signaling pathways of oncogenesis, tumor progression, and drug resistance. In mammals, MAPK proteins can be divided into several subgroups including extracellular signal-regulated kinases 1 & 2 (ERK1/2), p38 MAP kinases, and c-Jun amino-terminal kinases (JNK), and the family of ERK5 proteins and atypical ERK3, 4, 7, and 8 [27] (Figure 1). The activating of the EPO/MAPK signaling pathway includes the following steps: recruitment of SH2 inositol 5-phosphatase 1 (SHIP1) to EPOR and simultaneously binding adapter proteins such as the growth factor receptor-bound protein 2 (GRB2), son of sevenless (SOS), and SHC Adaptor Protein 1 (SHC1) [28,29]. Subsequently, RAS and RHO GTPases are involved in the activation of downstream MAPK members such as RAFs, MEKs, and ERKs. About 160 ERK1/2 substrates were identified [30] and their activation depends on particular extracellular and intracellular conditions leading to the appropriate response of the cell.

During erythroblast differentiation, an upstream signaling molecule of EPOR spleen tyrosine kinase (SYK) is constitutively associated with EPOR in the plasma membrane and is required for STAT5 and ERK activation induced by EPO [31]. Maintenance of human erythropoiesis is also regulated by synergic activation of EPOR and transferrin receptors 2 (TFR2), mediating signal via MAPK/ERK and PI3K/AKT [32]. TFR2, which is important for sustaining both iron delivery and signaling, associates with EPOR in the endoplasmic reticulum of erythroblasts, creating a complex transported to the cell surface [33]. During erythroblast development, other important EPOR interacting factors have been described such as the protein RHEX that has stage-specific expression, increases ERK1/2 activation, and is associated with EPOR. Other downstream EPOR/MAPK signal transducers include neurofibrin that maintains a balance during ERK1/2 signaling and RAS/MEK mediators MASL1 and Rasa3 [34]. Once activated, ERK1/2 is shifted into to the nucleus and phosphorylates various transcription factors such as proto-oncogenes c-Myc, c-Fos, c-Jun, ETS domain-containing protein Elk-1, and the cyclic AMP-dependent transcription factor ATF2. The activation of a particular set of target genes depends on extracellular and intracellular stimuli resulting in appropriate cellular response. The family of MAPK members is wide and may have opposite effects during signal transduction. While activation of p38 MAPK is associated with stress-induced erythropoiesis [35], ERK1 serves as a negative regulator of the steady-state splenic erythropoiesis [36] operating not through the EPO/EPOR signaling pathway but involving the Sonic Hedgehog/BMP4 pathway [37,38]. There are many negative regulators involved in the MAPK pathway including the Spred 1 protein, Raf kinase inhibitor protein RKIP, p38 MAPKs and JNKs downregulator DUSP1/MKP1 [39,40,41]. Cytoplasmic ERK1/2 can shape also a negative feedback regulatory mechanism upstream of the ERK pathway by phosphorylation of protein kinases SOS, Raf-1, and MEK [42].

## 4. The Interconnection between EPO Signaling Cascades

JAK2 kinase is crucial for successful EPO/EPOR activation of the PI3K and MAPK pathways. The pair of JAK2 molecules is constitutively associated with cytoplasmic box 1 on the intracytoplasmic site of EPOR [43]. After EPO binding to the EPOR, transphosphorylation of JAK2 kinases occurs and this further leads to phosphorylation of tyrosine residues of EPOR on which the STAT5, MAPK, and PI3K signaling pathways are activated. Although JAK2 is necessary for EPO signaling [44,45], the extent to which the MAPK and PI3K pathways are dependent on JAK2 phosphorylation has not yet been studied. However, JAK2 mutation in patients with myeloproliferative neoplasms (polycythemia vera) caused overactivation of the PI3K and MAPK pathways [46].

PI3K/AKT and MAPK pathways have numerous overlaps and cross-talks between each other to support mechanisms of cell survival. Indeed, they act as reciprocal inhibitors [47]. It is well known that the RAS protein, a joining molecule between MAPK and PI3K, activates RAF by triggering the MAPK/ERK pathway [48], similarly recruiting p110 catalytic subunit of PI3K to the plasma membrane and activating the AKT signaling pathway [49]. Signal transducers such as protein GAB1 joins PI3K-mediated EPO signals with the MAPK pathway [50] or a second lipid messenger PIP_2_ also mediates both AKT and MAPK signaling by activating protein kinase C (PKC) that mediates the signal to RAF [51]. Moreover, targeted inhibition of one pathway can manage the flow of PIP_2_ signals to another. PREX1, an important guanine nucleotide exchange factor, is also an activator of both AKT and MAPK signaling during tumorigenesis [52]. Interestingly, a compensatory effect as an adaptation to chemotherapy in tumors was described, whereafter attenuating MAPK signaling AKT emerged to be activated much stronger [53].

## 5. EPOR/PI3K/AKT and EPOR/MAPK in Non-Hematopoietic Tissues

EPO signaling cascades are activated in many non-hematopoietic tissues in order to prevent against tissue injury and damage (Figure 2). Either endogenous EPO production or the expression of EPOR provide an auto/paracrine loop with its protective and anti-apoptotic effect. EPO mediated effects on the cell depend on many factors such as cell type, the type of EPO receptor present on the cell surface, and physiological/patophysiological conditions, among other factors. Despite the fact that erythroid and non-erythroid EPO/EPOR signaling share many similarities, there are some important differences. In addition to classical full length EPO, non-erythropoietic splice variants of human and murine EPO have been detected. Among them, EV-3 variant without exon 3 present in human serum exerted a protective effect on neuronal cells [54]. Furthermore, the β-common receptor (βcR) known as CD131 and Cytokine receptor like-factor 3 (CRLF3) as an alternative EPOR exists in non-erythroid tissues [55]. Signaling through these receptors and/or receptor complexes demonstrate distinct biological outcomes, while the molecular mechanism of such signaling is mainly unknown. For example, the study of He et al. [56] revealed that in contrast with the active EPOR homodimer, the EPOR/βcR complex does not utilize the cytoplasmic tyrosines for EPOR signaling.

### 5.1. Mitochondria

The EPOR/PI3K pathway operating in the nucleus is well described in erythroid progenitors, while non-hematopoietic tissues also use mitochondrial engagement. It is well known that the protection of the mitochondrial membrane integrity and the stopping of cytochrome c release from the mitochondria is the main mechanism by which AKT prevents cell death [58]. Moreover, mitochondrial biogenesis is one of the cardioprotective effects during hypoxia and it is regulated through the EPO/AKT/eNOS signaling pathway [59]. It has been proven that either genetic or pharmacological activation of AKT by EPO modulates mitochondrial morphology in which apparent mitochondrial elongation was seen to be dependent on the AKT activated Mitofusin-1 [60]. EPO induces also a complex formation of activated AKT and adenine nucleotide translocase, a major subunit of mitochondrial permeability transition pore (ANT), leading to the elevation of the threshold for its opening [61]. Interestingly, for EPO, antiapoptotic impact WNT1 and FOXO3a proteins are required to control mitochondrial membrane depolarization, cytochrome c release, and caspase activation [62].

### 5.2. Nervous System

Carelli et al. [63] discovered that increased endogenous brain EPO is associated with preventing ischemia/reperfusion damage of brain tissue. Indeed, therapeutic use of EPO is a promising neuroprotective medication used against hypoxia/ischemia in pre-term babies [64]. EPO together with IGF-I exerts cooperative initiation of neuroprotection via activation of the PI3K/AKT pathway [65]. Novel members associated with EPOR/PI3K/AKT axis such as the FAIM2 and GRINA factors have been discovered [66,67], approving its neuroprotective effect. Indeed, EPO acts as a guard in the nervous system via stimulation of the PI3K/AKT/GSK-3β pathway [68,69], AKT/mTOR/p70S6K pathway [70], and through the activation of PI3K/AKT/FOXO3a signaling [71]. During erythropoiesis, homodimeric EPOR is involved but in the nervous system, EPO acts upon a heteroreceptor complex comprising both the EPOR and βcR. Experiments with an EPO derivative carbamylated erythropoietin (CEPO) unlike EPO demonstrated that signal response did not act via JAK2 but through the AKT activated by the glial-derived neurotrophic factor (GDNF) [72,73]. For neuroprotective activity in hippocampal neurons, AKT signaling is essential regardless of the presence of functional STAT5. Conversely, to achieve the neurotrophic effect of EPO, the simultaneous activity of both STAT5 and AKT is required [74].

Diverse cross-talks between EPO-mediated MAPK and PI3K/AKT signaling exist in the nervous system and their complex interaction depends on the respective cellular process and the cell type. The synergistic effect of ERK1/2 and AKT activation by EPO was required for neuroprotection [75] and for EPO-induced phrenic motor facilitation in the nervous system [76]. Conversely, EPO protects neurons against apoptosis induced by oxygen and glucose deprivation that is closely related to the activation of PI3K/AKT and inactivation of the ERK1/2 signaling pathway [77]. Moreover, the protective effect of EPO mediated by ERK signaling in the case of disrupted the PI3K/AKT pathway proves a compensatory mechanism ongoing in impaired cells in which one pathway substitutes another to prevent cell damage [78]. The EPO mediated MAPK and PI3K pathways have an important impact on neuronal tissues, transducing the signal either to the neuroprotective Nrf2/Are pathway resulting in the reduction of oxidative stress [79,80] or to NF-κB axis promoting differentiation of neuronal stem cells into astrocytes [81]. Recently, the MAPK/ERK pathway was identified as a downstream effector of the EPO signaling pathway for migration and positioning of neurons in the developing neocortex [82,83]. On the contrary, in Schwann cells, EPO inhibited the microglial MAPK pathway to maintain myelin integrity [84]. Indeed, the neuroprotective effect of the ERK pathway is associated with either its activation [85] or inhibition [86] and depends on factors such as cell lineage, culture environment, and pathological circumstances. In this regard, short-term activation of ERK by growth factors under a physiologic condition is associated with neuroprotection, while prolonged and persistent ERK activation after injury may induce cell death [85]. Trophic factors such as EPO or BDNF might not even initiate transient ERK activation but attenuate the pathological long-lasting ERK activation after hypoxic injury and improve cell survival [87].

### 5.3. Bone and Bone Marrow

It is known that endogenous EPO regulates the bone environment on diverse levels. In bone, EPOR receptors are present on osteoblasts, osteoclasts, and on bone marrow stem cells (BMSCs) that can differentiate into osteoblasts, chondrocytes, and bone marrow adipocytes [57]. During the process of differentiation in osteoclasts, EPOR expression gradually decreases [88].

Regulation of the bone marrow microenvironment is associated with the EPO/MAPK pathway activity, inhibiting the adipogenic differentiation of bone marrow mesenchymal stem cells [89]. Interestingly, mice lacking endogenous EPO signaling have increased marrow adipogenesis and reduced ectopic bone formation [90]. The proliferation and migration ability of bone marrow-derived mesenchymal stem cells is significantly influenced by the EPO-regulated MAPK and PI3K/AKT signaling pathways [91] likely via the mechanism of increasing SDF protein chemokine molecule, an important player in directing the migration of stem cells. In addition, an intramuscular injection of EPO resulted in BMSCs mobilization to bone damage, increased bone regeneration process, and improved bone strength [92]. EPO/MAPK and EPO/PI3K signals are also involved in the protection of bone marrow microvascular endothelial cells in which nitric oxide (NO) donor increased the expression of EPOR through the activated MAPK under both normoxia and hypoxia conditions. On the contrary, EPO did not increase MAPK activity while it induced AKT phosphorylation in endothelial cells under both normoxia and hypoxia conditions [93].

### 5.4. Heart

EPO’s protective effect was described in a wide variety of processes in cardiovascular pathophysiology, particularly in ischaemia, cell proliferation, apoptosis, and platelet activation. EPO exhibited myocardial protection through the MAPK/ERK induced GATA-4 stability, reduction of caspase-3 activity, upregulation of Bcl-2 [94], and decreased myocardial fibrosis via suppressing the NADPH/ERK/NF-κB pathway [95]. Paracrine EPO accelerated the healing process in intramyocardial cardiac angiogenetic stem cells via the activation of AKT signaling and through the upregulation of upstream signals FOS and FZD7 in addition to the activation of TGF-β/WNT signaling [96]. Moreover, anti-inflammatory and antifibrotic activity of EPO were demonstrated acting via both PI3K/AKT signaling and downregulating Toll-like receptor (TLR4) expression in addition to inhibiting the release of TGF-β1, TNFα, IL-6, IL-1β, IL-17A, MMP-9, and MMP-2 factors [97].

### 5.5. Kidney

EPO and its derivatives such as helix B surface peptide (HBSP) revealed protective effects in transplant-related renal injuries such as ischemia-reperfusion injury (IRI) and immunosuppressant nephrotoxicity [98]. Renoprotection of EPO was demonstrated via modulation of the STAT6/MAPK/NF-κB pathway, ERK/p53 signaling [99,100], and via the PI3K/AKT activated NOS/NO pathway [101,102].

### 5.6. Muscles

The importance of EPO in the proliferation, migration, and invasion via p38 MAPK and ERK1/2 signaling was confirmed on vascular smooth muscle cells [103]. Furthermore, EPO promoted proliferation, survival, and wound recovery in myoblasts via the PI3K/AKT pathway [104]. On the contrary, a single EPO injection during exercise was not sufficient to trigger the PI3K/AKT pathway, the main positive regulator of muscle protein synthesis in skeletal muscles [105]. Nevertheless, EPO may be considered as an effective practical therapeutic option for muscle-injury recovery [106,107].

### 5.7. Retina

The expression of EPO and EPOR in retinal tissues indicates its autocrine or paracrine action. EPO can protect retinal tissues including microvascular endothelium, retinal epithelium, Müller cells, and retinal neurons. This effect of EPO is partly dependent on the activation of PI3K and MAPK pathways. During diabetic retinopathy, inner blood-retinal barrier (BRB) breakdown occurs while EPO prevents this damage via the inhibition of microglial activation and phagocytosis mediated by the SRC/AKT/COFFILIN signaling pathway [108]. Furthermore, BRB integrity was maintained by EPO-regulated downregulation of HIF-1α, JNK signaling, and thus by the up-regulation of ZO-1 and occludin expression [109]. Another mechanism of retinal cytoprotection via maintaining zinc homeostasis by the EPOR/MAPK ERK pathway and upregulation of ZnT8 expression was also demonstrated [110]. Therefore, EPO can be considered a therapeutic tool for the treatment of diabetic macular oedema during diabetic retinopathy.

## 6. EPOR/PI3K/AKT and EPOR/MAPK in Cancer

Research progress in the area of erythropoiesis with extensive proteomics, transcriptomic, and epigenetic studies on EPO signaling is currently absent regarding malignant cells. Nevertheless, we can conclude that EPO signaling has several different features in cancer cells compared to erythroid tissue. Firstly, cancer cells have a lower number of receptors for EPO compared to erythroid progenitors [111]. Indeed, EPORs of cancer cells are located either on the membrane surface or as a soluble intracellular forms [112], and apart from the classical EPOR, ephrin-type B receptor 4 (EPHB4) as an alternative EPO binding form was identified in malignant cells [113].

In addition, microenvironments of bone marrow and tumors differ from each other, therefore EPO signaling may be influenced by distinct mechanisms and interactions of signaling molecules. In tumor tissues, autocrine or paracrine secretion of EPO is typical compared to endocrine distribution of EPO required for erythropoiesis. Moreover, the hypoxic microenvironment of cancer tissues stimulates the local EPO production, supporting viability and independent tumor growth. While EPO signaling during erythropoiesis plays a main role in differentiation and proliferation, in cancer cells the signaling refers more to anti-apoptotic cell survival and migration activity. Importantly, the main framework of EPO signaling remains unchanged across all types of cells.

EPOR and its downstream signaling pathways STAT5, PI3K/AKT, and MAPK were found as constitutively active and independent of EPO stimulation in non-small cell lung, renal, and ovarian cancer cells but the mechanism underlying this phenomenon is not sufficiently described [114,115,116]. Nevertheless, one of the mechanisms explaining the independence of the EPO could be a mutation of the KRAS gene [117]. In addition, after the downregulation of EPOR, both phosphorylated EPOR and STAT5 levels significantly decreased and either pAKT in ovarian carcinoma cells or pERK1/2 in human renal carcinoma cells were increased as a compensatory mechanism [114,115].

EPO activates the PI3K/AKT pathway in human melanoma cells [118] and both the antiapoptotic and proliferative effect of EPO acting through the AKT and ERK pathways in neuroblastoma cancer cells were observed [119]. Moreover, enhanced EPO and EPOR expression with AKT signalization included resulted in the promotion of colon cancer cell growth, proliferation, and angiogenesis, and the same applies in the EPO-promoted proliferation of glioma cells in vitro and in vivo [120,121]. Additionally, EPO significantly enhanced the proliferation of rat pancreatic tumor cell line AR42J via activation of ERK1/2 and JNK1/2 [122]. MAPKs together with the STAT5 and AKT signaling pathways are necessary also for EPO-mediated antiapoptotic effect in differentiated neuroblastoma SH-SY5Y cells [123]. This study elucidated that the activation of a single MAPK signaling pathway per se was not sufficient for antiapoptotic activity, thus STAT5 and AKT axis must also be included [123]. On the contrary, EPO-induced activation of the MAPK and PI3K/AKT pathways were sufficient for growth support and for the protection of human breast cancer cell lines against apoptosis [124]. While functional EPOR signaling was essential for the breast tumor progression effect of EPO and emphasizes the importance of the EPO/EPOR axis, tumor growth was markedly reduced after the knockdown of EPOR. Chan et al. (2017) also confirmed that EPO-induced MYC expression was mediated through the MAPK and PI3K/AKT pathways [124]. Furthermore, pathologically deregulated erythropoiesis (polycythemia vera) is associated with an abnormal increase in the activation of the MAPK and PI3K/AKT pathways [46] likely caused by the point mutation in JAK2 kinase.

Cell migration is considered an initial step in metastasis and may be regulated by a variety of signaling pathways. Interestingly, the migration of human breast cancer cells with stable overexpression of EPOR was promoted by increased activation of the ERK1/2 pathway but not through the JAK2-STAT5 axis [125]. On the contrary, EPO in a JAK-dependent manner enhanced cell migration and activated RhoA protein via MAPKs an in EPOR-expressed cervical cancer cell line [126]. Co-signaling of EPO and the stem cell factor (SCF) activated both ERK1/2 and JAK2/STAT5, and had a cooperative effect on the migration ability of cervical cancer cells [127]. Whereas SCF and EPO/SCF induced strong, sustained phosphorylation of ERK1/2, EPO solely induced only a modest, transient activation of ERK1/2 in cervical cancer cells. Indeed, the results of Aguilar et al. (2014) demonstrate the cooperative activity of EPO and SCF in cells, expressing their cognate receptors followed by the co-signaling of two cytokine receptors, induced migratory behavior, and anchorage-independent cell growth. It was also recorded that constitutively activated EPOR selectively induced MEK/ERK members of the MAPK pathway but not p38 MAPK mediators [128]. Interestingly, the EPO-induced activation of the PI3K/AKT and MAPK/ERK pathways was reduced using the hematopoietic cell kinase (HCK) inhibitor (iHCK-37) particularly in the cells with high HCK expression [129]. During tumorigenesis, the MAPK and PI3K pathways may interact also with many other pathways such as NOTCH, AR, WNT/β-catenin, TGF-β, COX-2, and so on, resulting in an increase of cancer aggressiveness [130,131]. Nevertheless, further research must be performed to prove the contribution of AKT, MAPK, and their cooperation regarding the EPO/EPOR or constitutive EPOR in cancer cells.

## 7. Conclusions

The interaction of EPO and EPOR triggers the activation of several signaling pathways including PI3K/AKT and MAPK. They are known to be important regulators of the differentiation, proliferation, and cell survival of erythroid cells. More information is emerging about their role in the protection of non-hematopoietic tissues and cancer cells. In this article, we have attempted to summarize recent findings on the EPO-induced PI3K and MAPK pathways, and described their role in numerous cellular events. Currently, while the entire process of erythroid differentiation is influenced by many different interventions, more investigations are required to uncover the diverse effects of EPO on the erythropoiesis in addition to its well-characterized proliferative and pro-survival effects. The future challenge in this area is to intersect detailed interactions of signaling molecules and determine their contribution to and beyond erythropoiesis. Advances in this area of research may be useful for improving the efficacy of EPO therapy in clinical use and to gain a better understanding the overall processes of cellular signaling.

## Figures and Tables

**Figure 1 ijms-22-07682-f001:**
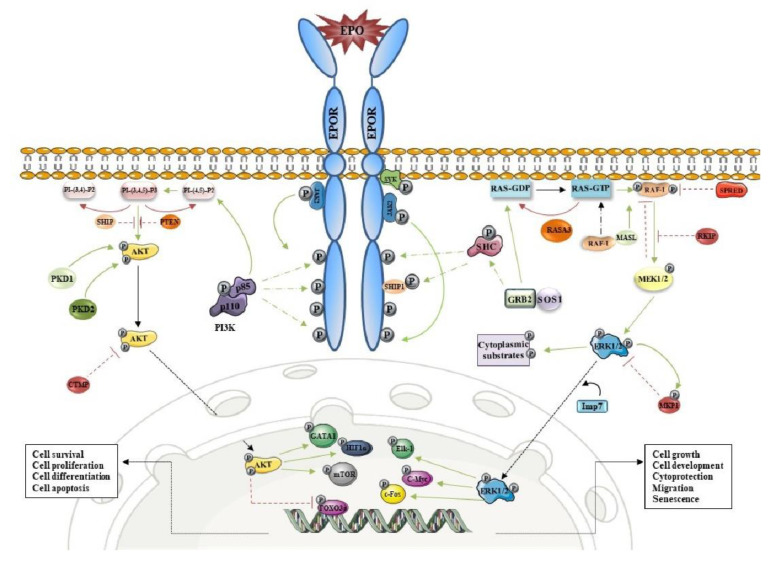
The main mediators of the EPOR/PI3K and EPOR/MAPK signaling pathways with cell response to their activation. Green line: activation effect (phosphorylation); red line: inhibition effect; and black line: transition.

**Figure 2 ijms-22-07682-f002:**
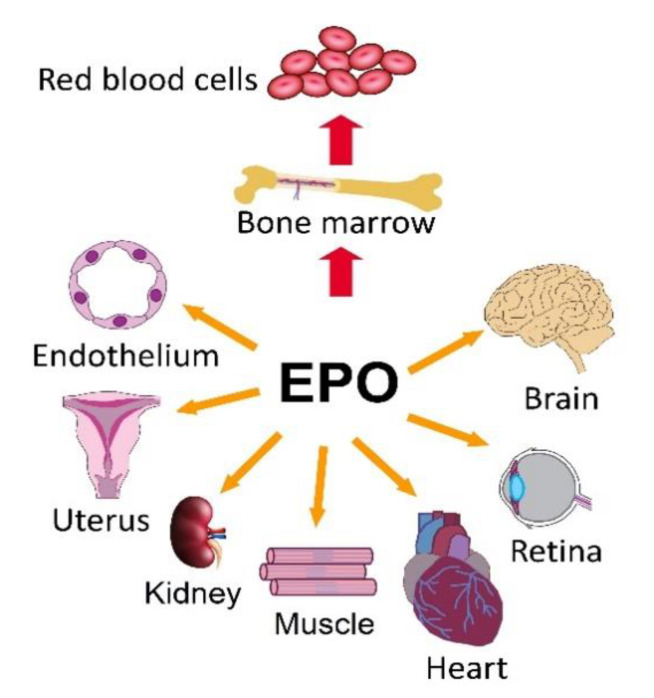
Non-hematopoietic sites of EPO/EPOR. Adapted with permission from ref. [57]. Copyright year 2020, owner’s name Constance T. Noguchi.

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
