# Peer review of "The Role of PI3K/AKT and MAPK Signaling Pathways in Erythropoietin Signalization"

_ijms, 2021, doi:10.3390/ijms22147682_

Round 1
Reviewer 1 Report
This review focuses on MAPK and PI3K/AKT signaling pathways triggered by EPO highlighting new clinically relevant modulators and EPOR expression. The aim of the authors is to describe the influence of signaling pathways in hematopoietic, non-hematopoietic tissues and tumor cells. However it is not a critical evaluation of the emerging data but a long list of topics, hard to follow, sometimes copying sentences from the abstract of referenced papers. Furthermore, the authors do not highlight the differences between EPO signal pathways in erythropoietic and non-erythropoietic cells as referred to in the abstract.
It is not clear if the new findings are referred to animal models or are translated on human-trials.
EPO pathways in non-hematopoietic tissues should be reported before the section on EPO pathways and cancer.
I suggest to remove reference 52 since it is not acceptable to report a Chinese language reference in an international manuscript.
The manuscript is poorly written, and need to be reworked.
Author Response
Comments and Suggestions for Authors
This review focuses on MAPK and PI3K/AKT signaling pathways triggered by EPO highlighting new clinically relevant modulators and EPOR expression. The aim of the authors is to describe the influence of signaling pathways in hematopoietic, non-hematopoietic tissues and tumor cells.
1.However it is not a critical evaluation of the emerging data but a long list of topics, hard to follow, sometimes copying sentences from the abstract of referenced papers.
Dear reviewer, thank you for your compelling comments on our review paper.
In the last ten days, we tried to rework our review in the best possible way.
We have organized clear arrangement of the chapters starting from description of general erythropoiesis, throughtout EPO/PI3K/MAPK contribution to erythropoiesis, followed by the chapter of EPO signaling in non-hematopoietic tissues and finished with the EPO axis in cancerous tissues. Regarding copying of sentences from another papers, test of the originality proves our primary intellectual contribution. Lastly, we also paid attention not repeating sentences in the text as you can see.
2.Furthermore, the authors do not highlight the differences between EPO signal pathways in erythropoietic and non-erythropoietic cells as referred to in the abstract.
For better understanding, we mention specific differences between particular tissues at the beginning of non-hematopoietic and cancer chapters.
3.It is not clear if the new findings are referred to animal models or are translated on human-trials.
The most studies on EPO signalisation are performed on human cells, which is mentioned in the text. There are few references reffered to in vivo expreriments, which we hope are clearly shown in the text. One way, how to solve your question is to scroll through the text and specify study after study, so please consider if this is necessary.
4.EPO pathways in non-hematopoietic tissues should be reported before the section on EPO pathways and cancer. Thank you for your suggestion, we have already corrected.
- I suggest to remove reference 52 since it is not acceptable to report a Chinese language reference in an international manuscript. Thank you for suggestion, we have already corrected.
Reviewer 2 Report
In this review paper, Tóthová et al discussed. The roles of EPO-regulated MAPK and PI3K/AKT signaling pathways in erythropoiesis and in other cellular events in non-erythroid cells. The manuscript is well-organized and well written. This reviewer only has some minor concerns.
Specific concerns:
- In the title, “Pathwats” should be “Pathways”.
- In page 2, line 81, “pluripotent” might be “multipotent”?
- This sentence in page 3, “Interaction of EPO and its receptor EPOR leads to activation of JAK2/STAT5, PI3K/AKT and MAPK signaling pathways”, is redundant, as it’s already introduced in Section 1.
- Section 5.4, “Hearth”, should be “Heart”?
- With respect to the interplay between MAPK and PI3K/AKT pathways, the authors only discussed briefly in the section 3. Does EPO-EPOR interaction activate the three pathways simultaneously in the same cell? Or does it activate each pathway in a cell type- or context-specific way? It will be very helpful if the authors can clarify such questions in the section 3.
- With respect to the role of EPO-EPOR in non-erythroid cells, does EPO exert its effect absolutely through EPOR? How is the tissue distribution of EPOR expression pattern?
Author Response
In this review paper, Tóthová et al discussed. The roles of EPO-regulated MAPK and PI3K/AKT signaling pathways in erythropoiesis and in other cellular events in non-erythroid cells. The manuscript is well-organized and well written. This reviewer only has some minor concerns.
Specific concerns:
- In the title, “Pathwats” should be “Pathways”. We have already corrected.
- In page 2, line 81, “pluripotent” might be “multipotent”? We have already corrected
- This sentence in page 3, “Interaction of EPO and its receptor EPOR leads to activation of JAK2/STAT5, PI3K/AKT and MAPK signaling pathways”, is redundant, as it’s already introduced in Section 1. We have already erased.
- Section 5.4, “Hearth”, should be “Heart”? We have already corrected.
- With respect to the interplay between MAPK and PI3K/AKT pathways, the authors only discussed briefly in the section 3. Does EPO-EPOR interaction activate the three pathways simultaneously in the same cell? Or does it activate each pathway in a cell type- or context-specific way? It will be very helpful if the authors can clarify such questions in the section 3. According to your suggestion, we have described the cooperation of three main EPO signaling pathways and highlight the differences in EPO signaling among various cell types.
- With respect to the role of EPO-EPOR in non-erythroid cells, does EPO exert its effect absolutely through EPOR? How is the tissue distribution of EPOR expression pattern?
We have added alternative EPO receptors into the manuscript and prepared new figure showing EPO/EPOR expression in non-hematopoetic tissues.
Reviewer 3 Report
The authors summarized functional pathways (PI3K/AKT and MAPK) regulating erythropoiesis in normal and malignant cells. The manuscript is appropriately organized. However a number of improvements should be undertaken.
- The title contains typos. The suggestion is to change completely the title and make it general in a way that the authors will be able to describe comprehensively the erythropoietic process rather than focusing only on the PI3K/AKT and MAPK axis.
- Figure 1 and 2 should be combined in one and the font enlarged.
- A clear paragraph on erythropoiesis in normal physiology should be included.
- A paragraph describing hallmark diseases characterized by improper erythropoiesis needs to be added.
- Other pathways regulating erythropoiesis needs to be added in a separated paragraph.
- The addition of a figure depicting all organs in the body involved in the erythropoietic pathway could be added to diversify the two figures now present which are overlapping.
Author Response
Comments and Suggestions for Authors
The authors summarized functional pathways (PI3K/AKT and MAPK) regulating erythropoiesis in normal and malignant cells. The manuscript is appropriately organized. However a number of improvements should be undertaken.
- The title contains The suggestion is to change completely the title and make it general in a way that the authors will be able to describe comprehensively the erythropoietic process rather than focusing only on the PI3K/AKT and MAPK axis. We have corrected the title.
- Figure 1 and 2 should be combined in one and the font enlarged. Thank you for recommendation, we have combined figures.
- A clear paragraph on erythropoiesis in normal physiology should be included. According to your suggestion, we have added new chapter describing general erythropoiesis and mainly EPO effects.
- A paragraph describing hallmark diseases characterized by improper erythropoiesis needs to be added. We considered your suggestion, but we concluded that with respect to broad area of research, this topic deserves separately written review paper.
- Other pathways regulating erythropoiesis needs to be added in a separated paragraph. We have recenlty published a paper about STAT5 protein and its role in EPO signalization, so we decided to just quote this paper at the and of the chapter “EPO and erythropoiesis“.
- The addition of a figure depicting all organs in the body involved in the erythropoietic pathway could be added to diversify the two figures now present which are overlapping. We have added such a figure.
Round 2
Reviewer 3 Report
The authors have answered all the queries.